# Fetal temporal sulcus depth asymmetry has prognostic value for language development

Lisa Bartha-Doering [1✉], Kathrin Kollndorfer [1,2], Ernst Schwartz[3], Florian Ph.S. Fischmeister [2,4,5], Georg Langs [3], Michael Weber[2], Sonja Lackner-Schmelz[1,2], Patric Kienast[2], Marlene Stümpflen [2], Athena Taymourtash[3], Sophie Mandl[1], Johanna Alexopoulos[1,6], Daniela Prayer[2], Rainer Seidl[1] & Gregor Kasprian[2]

In most humans, the superior temporal sulcus (STS) shows a rightward depth asymmetry. This asymmetry can not only be observed in adults, but is already recognizable in the fetal brain. As the STS lies adjacent to brain areas important for language, STS depth asymmetry may represent an anatomical marker for language abilities. This study investigated the prognostic value of STS depth asymmetry in healthy fetuses for later language abilities, language localization, and language-related white matter tracts. Less right lateralization of the fetal STS depth was significantly associated with better verbal abilities, with fetal STS depth asymmetry explaining more than 40% of variance in verbal skills 6–13 years later. Furthermore, less right fetal STS depth asymmetry correlated with increased left language localization during childhood. We hypothesize that earlier and/or more localized fetal development of the left temporal cortex is accompanied by an earlier development of the left STS and is favorable for early language learning. If the findings of this pilot study hold true in larger samples of healthy children and in different clinical populations, fetal STS asymmetry has the potential to become a diagnostic biomarker of the maturity and integrity of neural correlates of language.

[1] Department of Pediatrics and Adolescent Medicine, Comprehensive Center for Pediatrics, Medical University of Vienna, Vienna, Austria. [2] Division of Neuroradiology and Muscoskeletal Radiology, Department of Biomedical Imaging and Image-guided Therapy, Medical University of Vienna, Vienna, Austria. [3] Computational Imaging Research Lab, Department of Biomedical Imaging and Image-guided Therapy, Medical University of Vienna, Vienna, Austria. [4] Institute of Psychology, University of Graz, Graz, Austria. [5] BioTechMed-Graz, Graz, Austria. [6] Department of Psychoanalysis and Psychotherapy, Medical University of Vienna, Vienna, Austria. ✉email: elisabeth.bartha-doering@meduniwien.ac.at

Although most organisms have symmetrical bodies, some species show structural and functional brain asymmetries. This is true for fish, amphibians, reptiles, and birds, which show functional laterality in escape behavior, eye use, rotational swimming, or aggressive responses[1]. In mammals like rats and cats, certain regions of the cerebral cortex are significantly thicker in the right hemisphere than in the left[2,3]. Furthermore, primates exhibit leftward structural asymmetries in the planum temporale and the Sylvian fissure[4]. Analogously, most adult human brains have a larger planum temporale and a longer Sylvian fissure in the left hemisphere[5]. Only humans, however, show a longer left and deeper right superior temporal sulcus (STS) with substantial cytoarchitectonic asymmetries in these regions[6,7]. STS asymmetry has a polygenetic background and appears to be independent from mechanisms determining laterality of other organs[8–10]. Moreover, the characteristic pattern of temporal lobe asymmetry seems to be most easily recognizable during prenatal brain development. Early post mortem anatomical studies as well as in vivo fetal MR imaging analyses have found that the right STS develops 1–2 weeks earlier than the left STS and that it is significantly deeper in over 90% of fetuses[11–14]. Lateralization of the STS is supposed to represent an anatomical marker associated with a highly relevant cognitive function that is unique to human beings: the human language[14–16].

The STS is one of the longest sulci of the brain and occupies an important part of the temporal cortex. It is formed by the adjacent superior and middle temporal gyri and reaches from the temporal pole to the angular gyrus. Functional neuroimaging data suggest that the area around the left STS is responsible for speech perception, phonological, and semantic language processing, while the right STS is more involved in social cognition[17].

The identification of unique neural features that might predict language development is not only of scientific interest to explain the cognitive success of the human species[16], but also of growing importance in clinical settings such as fetal counseling. While cross-sectional studies exist that link structural temporal lobe asymmetry and language functioning in healthy children[18] and in different patient groups with language deficits[19–22], it has remained unclear if the association of human brain asymmetry and language abilities would be supported by a unique longitudinal study design following individuals from the earliest time point of phenotypic expression of human temporal lobe asymmetries—the second trimester of pregnancy—to childhood.

Based on the robustness of structural STS depth asymmetries at prenatal stages of brain development, this longitudinal study evaluated their impact on childhood language function, language localization, and language-related white matter tract organization in a cohort of healthy individuals.

## Results

**STS depth quantifications.** The overall study group consisted of 38 children (age 6–13 years, mean 8.85, SD 1.98, 14 girls) whose mothers were transferred to fetal MRI diagnostics due to clinical reasons and whose fetal MRIs were subsequently diagnosed as normal. Reasons for referral to fetal MRI have been either diseases of the mother including metabolic diseases ($n = 9$), amniotic fluid problems ($n = 3$), tumors ($n = 2$), and elevated risks of abortion (e.g., uterus rupture in previous pregnancy, previos abortion, $n = 6$), or suspected diseases of the fetus including urogenital abnormalities (e.g., kidney cysts, hydronephrosis, $n = 8$), gastrointestinal abnormalities (e.g., diaphragmatic hernia, gastrochisis, $n = 6$), club foot ($n = 2$), and brain abnormalities (1 plexus cyst, 1 mild ventriculomegaly). In all children, a neuropsychological examination, a structural magnetic resonance imaging (MRI) of the brain, a functional MRI (fMRI) for language localization, and diffusion tensor imaging (DTI) to identify language-related white matter bundles were performed. Fetal structural MRI and structural MRI at test revealed normal findings in all children. In the retrospective analysis of the fetal MRI, the STS could only be detected from the gestational age of 25 weeks (postmenstrual age) onward and quantified in 29/38 fetuses. In 9 children (gestational ages of 20–25 weeks), the STS was not visible yet. Thus, 29 study participants were included in the quantitative analyses of the fetal STS and in all further statistical analyses. The group of all study participants and the group of study participants with visible fetal STS did not significantly differ with regard to variables on background information or later language abilities (see Supplementary Table 1 for a detailed statistical comparison of groups).

Within the study group with visible fetal STS ($n = 29$; age 6–13 years, mean 9.19, SD 2.03, 10 girls; Table 1), the mean depth of the right STS was significantly larger than its left counterpart ($t = 6.494$, $p < .001$). The group mean laterality index (LI) of the STS depths thus showed a right lateralization. In the individual STS measurements, the LIs of the STS depths ranged from −80.18 to 33.80 and were right-lateralized in 20 fetuses, bilateral in eight fetuses, and left lateralized in one fetus.

The individual depths of the left and right hemisphere STS significantly correlated with the gestational age at the time of the fetal MRI (right STS: $r = 0.645$, $p < 0.001$; left STS: $r = 0.677$, $p < 0.001$), indicating a symmetrical effect of gestational age on each side of STS depth and larger STS depths in older fetuses (Supplementary Table 2). However, the gestational age at fetal MRI did not correlate with the LI of the fetal STS depth ($r = 0.12$, $p = 0.551$), thus, increasing gestational age at fetal MRI was not associated with a significant change in the asymmetry of the fetal STS depth (please see also Supplementary Fig. 1a–f).

Furthermore, measurements of the fetal STS depth (left STS depth, right STS depth, LI of the STS depth) did not correlate with the children's handedness or the families' socioeconomic status, nor with nonperceptual reasoning, all of which were evaluated at test. In addition, males and females did not significantly differ with regard to left STS depth, right STS depth, or LI of the STS depth (please see Supplementary Table 2 and Fig. 2 for more details on the statistical results involving background variables).

| Table 1 Study group with visible fetal STS ($n = 29$). | | |
|---|---|---|
| | **Mean (SD)** | **Range** |
| *Background information:* | | |
| Sex (f/m) | 10/19 | |
| Gestational age at fetal MRI (weeks) | 29.06 (2.58) | 25.29 to 35.43 |
| Age at test (years) | 9.10 (2.03) | 6.33 to 13.25 |
| Handedness at test (LI) | −0.67 (0.59) | −1.00 to 0.80 |
| Socioeconomic status at test (7-point-scale) | 5.18 (1.52) | 1.00 to 7.00 |
| Perceptual reasoning at test (z-score) | 0.07 (1.14) | −2.46 to 1.80 |
| *STS measurements:* | | |
| Right STS depth (mm) | 3.88 (2.06) | 0.90 to 10.70 |
| Left STS depth (mm) | 2.43 (2.05) | 0.20 to 9.00 |
| LI of the STS depths | −28.18 (22.55) | −80.18 to 33.80 |
| *Language examinations:* | | |
| Expressive vocabulary (z-score) | −0.15 (1.30) | −2.33 to 2.05 |
| Language comprehension (z-score) | 0.39 (0.97) | −1.64 to 1.88 |
| Verbal fluency (z-score) | −0.34 (0.83) | −2.05 to 1.15 |
| Verbal memory (z-score) | −0.13 (0.67) | −1.50 to 1.04 |

**Language abilities**. Language testing revealed typically developed verbal abilities in 24/29 children. Ten of these children showed above average abilities, while nine children revealed below average abilities in one or more language function. Five children exhibited reduced scores in one language function (four in expressive vocabulary, one in verbal fluency), but no child showed reduced abilities in more than one language domain; thus, no child exhibited a language impairment. Overall, language test results were heterogeneous and point to a wide distribution of language performances in the study group (see Supplementary Fig. 3 for a detailed figure of individual language abilities).

Language scores did not significantly correlate with any background variables, including age at test, handedness, and socioeconomic background (Supplementary Table 2). Furthermore, there was no sex difference with regard to language abilities. However, two language tests were significantly associated with nonverbal perceptual reasoning (expressive vocabulary: $r = 0.52$, $p = 0.004$; language comprehension: $r = 0.56$, $p = 0.001$).

**Language lateralization and localization**. The functional magnetic resonance imaging (fMRI) measurement of one child (m) had to be aborted due to claustrophobia, and measurements of five children (3m, 2f) had to be excluded due to excessive movement-related artifacts. Head movement of the remaining 23/29 children was within the tolerable limits (overall movement group mean 0.11 mm, SD 0.08, range 0.03–0.37).

In the fMRI group analysis, one-sample t-tests revealed a typical language localization pattern with left-lateralized activations in middle temporal regions including the insula, in the inferior frontal operculum, and in the superior frontal lobe. In the right hemisphere, the group of participants showed activations in the insula and the medial frontal lobe (Supplementary Fig. 4). In single-subject analyses, language was left lateralized in the overall cortex (LI total) in 17 children, bilateral in one child, and right lateralized in five children. Detailed information on LIs in different regions of interest in individual study participants and over the whole group is given in the Supplementary Information (Supplementary Table 3, Supplementary Fig. 5).

LIs of language lateralization did not significantly correlate with any background variables, including age at test, handedness, socioeconomic background, and perceptual reasoning (Supplementary Table 2). Furthermore, males and females did not significantly differ with regard to functional language lateralization.

**Language-related white matter tracts**. Data on white matter tracts are available in 25/29 children. Overall, fractional anisotropy measures revealed little hemispheric difference and was thus interpreted as bilateral in all single subjects, while LIs of fasciculi volumes and number of streams within the fasciculi showed more distributed values (Supplementary Table 5). Group analyses revealed a left lateralization of the arcuate fasciculus, the inferior longitudinal fasciculus, and in part of the superior longitudinal fasciculus I. Tracts I and II of the superior longitudinal fasciculus as well as the uncinate fasciculus showed a group mean LI towards the right hemisphere in volume and number of streams. Detailed information on LIs in individual study participants and over the whole group are given in the Supplementary Information (Supplementary Table 4, Supplementary Fig. 6).

LIs of language-related white matter tracts did not significantly correlate with any background variables, including age at test, handedness, socioeconomic background, and perceptual reasoning (Supplementary Table 2). Furthermore, boys and girls did not

**Table 2 LIs of the STS depths predict later language abilities.**

| Outcome | R* | p | Regression model equation |
|---|---|---|---|
| Expressive vocabulary | **0.43** | **0.007**\*\* | $y = 0.50 + 0.03x_1 + 0.60x_2$ |
| Language comprehension | 0.15 | 0.476 | $y = 0.53 + 0.01x_1 + 0.51x_2$ |
| Verbal fluency | **0.53** | **0.002**\*\* | $y = 0.14 + 0.02x_1 + 0.30x_2$ |
| Verbal memory | **0.59** | **0.000**\*\* | $y = 0.36 + 0.02x_1 + 0.22x_2$ |

Note: *correlation of LIs of the STS depths and language abilities, as indicated by multiple linear regression and adjusted for nonverbal perceptual reasoning ($n = 29$); $x_1$, LIs of the STS depths variable; $x_2$, nonverbal perceptual reasoning variable. Bold letters indicate $p < 0.05$, ** indicates significance after Bonferroni correction.

significantly differ with regard to language lateralization. However, boys tended to show more left lateralization of the superior fasciculus longitudinalis II, and children with higher socioeconomic background and better perceptual reasoning, respectively, tended to have more left lateralization of some tracts of the superior fasciculus longitudinalis.

**Prognostic value of fetal STS depth asymmetry for later language abilities**. Multiple linear regressions were carried out to test if the LI of fetal STS depth predicted language abilities 6–13 years later. Analyses were adjusted for nonverbal perceptual reasoning and showed that the model explained 45% of the variance in later expressive vocabulary ($F_{(2,26)} = 10.55$, $p < 0.001$, $R^2 = 0.45$), 43% of the variance in later verbal fluency ($F_{(2,26)} = 8.85$, $p = 0.001$, $R^2 = 0.43$), and 48% of the variance in later verbal memory ($F_{(2,26)} = 11.88$, $p < 0.001$, $R^2 = 0.48$), with less right lateralization of the fetal STS depths being significantly associated with better verbal abilities (Table 2, Fig. 1a–c). No associations were found between fetal STS depth asymmetry and later language comprehension. Tests to see if the data met the assumption of collinearity indicated that multicollinearity was not a concern (LI of STS depths, Tolerance $= 1.00$, VIF $= 1.00$; nonverbal perceptual reasoning, Tolerance $= 1.00$, VIF $= 1.00$). Furthermore, the data met the assumption of independent errors (expressive vocabulary, Durbin-Watson value $= 1.74$; verbal flexibility, Durbin-Watson value $= 1.54$; verbal memory, Durbin-Watson value $= 1.94$; language comprehension, Durbin-Watson value $= 1.93$).

**Association of fetal STS depth asymmetry and later language localization**. Nonparametric Spearman correlations showed no association of fetal STS depth asymmetry with later language *lateralization* in the overall brain (LI global R $= -0.28$, $p = 0.203$) nor with the language lateralization LIs of specific regions of interest (LI frontal R $= -0.13$, $p = 0.146$; LI temporal R $= -0.04$, $p = 0.848$; LI parietal R $= -0.106$, $p = 0.631$). However, second level multiple regression analyses of fMRI language *activations*, analyzed with SPM and adjusted for age at test and MR device used, revealed a significant association of the LIs of the fetal STS depth with activations in the left superior temporal lobe (Fig. 2). Thus, children with less right lateralization of the fetal STS depth showed significantly more left temporal language localization 6–13 years later.

**Association of fetal STS depth asymmetry and later language-related white matter tracts**. Multiple linear regressions, adjusted for age at test and MR device used, were carried out to test if the LI of fetal STS depth predicted the lateralization of language-related white matter tracts 6–13 years later. We did not include values of fractional anisotropy in the analyses due to their small

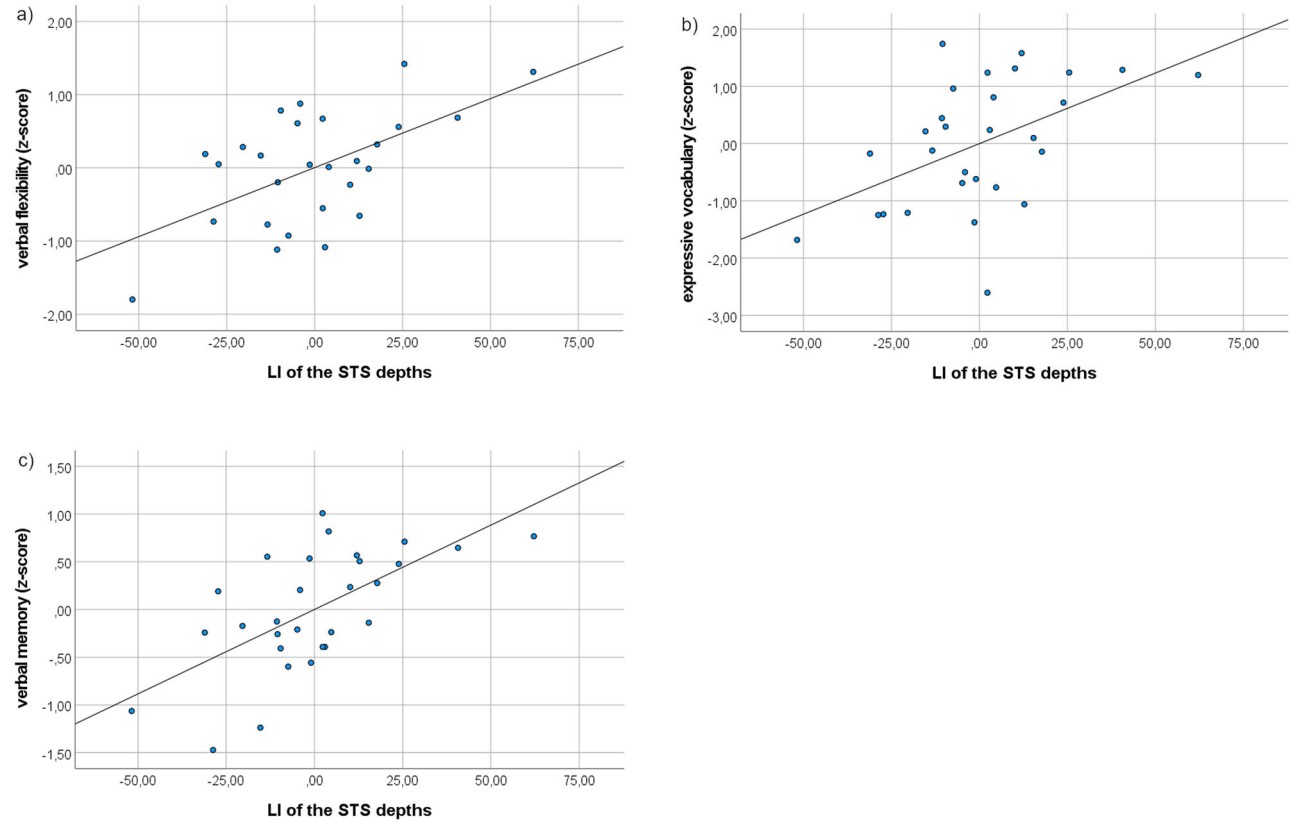

**Fig. 1 LIs of the fetal STS depths predict later. a** Verbal flexibility, **b** expressive vocabulary, and **c** verbal memory, indicated by multiple linear regression and adjusted for nonverbal perceptual reasoning ($n = 29$).

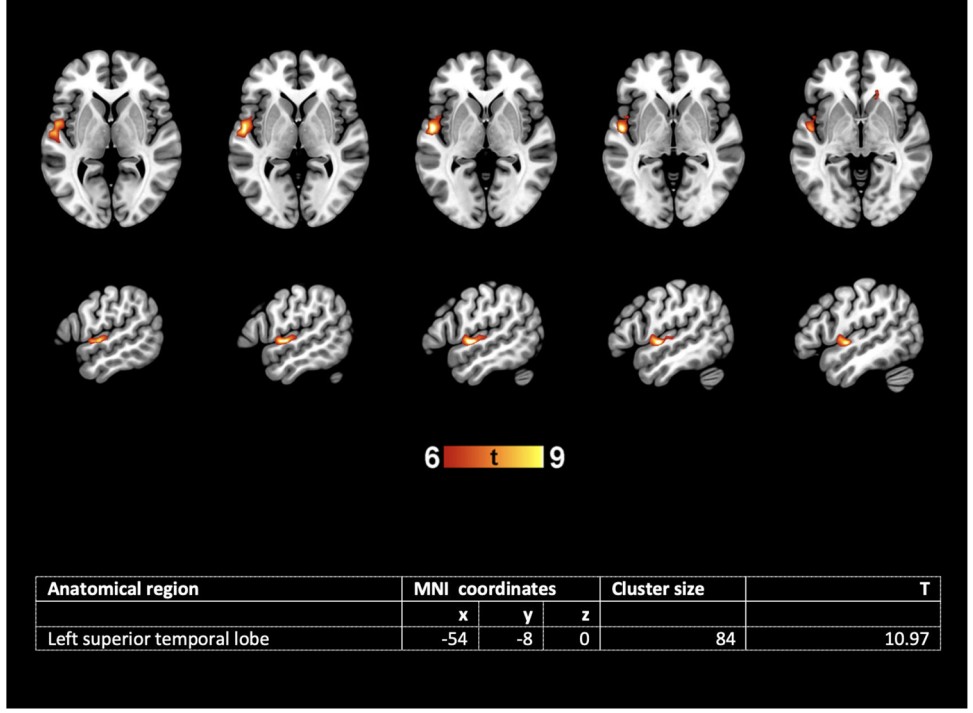

| Anatomical region | MNI coordinates | | | Cluster size | T |
|---|---|---|---|---|---|
| | x | y | z | | |
| Left superior temporal lobe | -54 | -8 | 0 | 84 | 10.97 |

**Fig. 2 LIs of fetal STS depths predict later language activation within the left superior temporal lobe (second level multiple linear regression analysis, adjusted for age at test and MR device; $n = 23$, $p < 0.05$, FWE-corrected).** Coordinates are given of the peak voxel in the activated cluster. Activation is depicted on a normalized T1 template, neurological convention.

variance. The linear regressions showed that fetal STS depth asymmetry did not predict any of the LIs of language-related white matter tracts (Supplementary Table 5). However, the regression model explained 41% of the variance in the volume of the left superior longitudinal fasciculus III ($F(3,21) = 4.95$, $p = 0.009$, $R^2 = 0.41$), with more left lateralization of the fetal STS depth being significantly associated with a greater volume of the left superior longitudinal fasciculus III. No other left or right fasciculus volume or stream amount was significantly correlated with fetal STS depth asymmetry.

## Discussion

The present study investigated the prognostic value of STS depth asymmetry in healthy fetuses for later language function, language localization, and language-related white matter tracts. Most fetuses in this study showed a deeper right STS compared to its left counterpart. Less right lateralization of the STS depth at fetal stages of brain development was, however, significantly associated with better verbal abilities, increased left temporal language localization, and a greater volume of the left superior longitudinal fasciculus III during childhood, between 6 and 13 years later.

STS depth asymmetry is a profound feature of a normally developed brain, as it is not only commonly observed in fetuses, but also in children, adolescents, and adults[16]. Indeed, the majority of healthy fetuses investigated in this study showed right lateralized STS depth asymmetry. Nevertheless, laterality indices of sulcal depth varied considerably, with those who exhibited less pronounced right sulcal depth lateralization at fetal age showing a significant larger vocabulary, better verbal fluency, and better verbal memory 6–13 years later. In fact, fetal STS depth asymmetry explained over 40% of variance in these verbal skills, when adjusted for nonverbal reasoning.

What drives fetal STS asymmetry? Individual STS asymmetry is rooted in a variety of genes associated with neuronal connectivity and migration[23], becoming phenotypically recognizable by modern prenatal imaging at around the 25th week of gestational age, and remaining present until adulthood. Like planum temporale asymmetry, it is not significantly influenced by sex or handedness[24]. However, we found a linear relationship of the structural fetal STS depth asymmetry with later language localization and language-related white matter tracts within the left temporal lobe. While a previous study did not find a significant difference in STS depth asymmetry between adults with left-versus right lateralized language localization during silent verb generation[16], a subsequent study in 95 adults demonstrated a significant association between STS depth and voice-sensitive functional activity and even predicted the anatomo-functional correspondence at the individual level[25]. Cross-sectional studies in adults cannot infer a causal link between STS depth asymmetry and functional localization of language-related areas; however, the present longitudinal study may shed more light on the development of this relationship.

The fetal auditory system is functional around the 25th weeks of gestational age[26], and its maturation is subsequently driven by environmental stimuli[27,28]. Functional language areas within the left hemisphere develop successively[29,30]. While auditory perception is processed in the STS bilaterally, most of the language-sensitive region is located in the depths of the left STS[17,25]. Early cortical folding is genetically driven and has been hypothesized to be closely associated with an optimal organization of functional areas and their white matter connections[31]. Sulcal depth is positively correlated with adjacent gyri volume and white matter connections[32,33]. A severe delay in gyrification development is predictive of the neurodevelopmental outcome of premature newborns at term-equivalent age[14]. A recent study also showed

that in very preterm born infants the curvature of the left superior temporal gyrus at term-equivalent age is associated with the neurodevelopmental outcome at two years of age[34]. Thus, we hypothesize that an earlier and/or more localized functional and structural development of language-related areas within the left temporal cortex during fetal age is accompanied by an earlier development of the left STS, thus inducing less right STS depth lateralization, and is favorable for early language learning.

Atypical patterns of the STS and altered structural temporal asymmetries have been described in various neurodevelopmental diseases defined by or highly associated with language deficits. In autism, a significant body of neuroanatomical studies exhibits STS abnormalities (for review, see e.g., Redcay et al.[35]). Furthermore, children with developmental dyslexia show atypical sulcal patterns in left temporal areas[36]. In addition, children with specific language impairment present less gray matter in the STS than typically developing children[37]. Remarkably, reduced leftward language lateralization has been described in all these patient groups[22]. These structural alterations in sulcal depth may indicate (genetically-driven) early deficits in neuronal proliferation, migration, and/or pruning in the STS during fetal stage, which result in atypical functional language organization and reduced language abilities.

In older children and adults, STS depth asymmetry has been explained in part by the larger number of sulcal interruptions in the left than in the right hemisphere in both children and adults[16]. These interruptions, also called *pli de passage*, develop and increase in the STS between birth and two years of age[38], while there are no such STS interruptions in early fetal development. However, these pli de passage are thought to be more or less superficial across individuals and hemispheres and may thus be an important source of variability in sulcal depth[25]. This possibly explains why fetal STS asymmetry may be a more sensible measure for language development than STS depth measures later in life and may provide an early window into the pathways to language.

While STS depth asymmetry predicted later expressive vocabulary, verbal fluency, and verbal memory, the present study did not find an association between early STS depth asymmetry and later language comprehension. This is surprising considering the fact that perception of language stimuli is probably the first linguistic act of fetuses and newborns. Therefore, earlier left-sided language areas might enhance language comprehension. However, many lesion-symptom correlation studies have highlighted several brain regions outside the left temporal lobe that contribute to language comprehension, and functional imaging studies have shown distributed activations patterns during the performance of tasks involving language comprehension[39]. One might hence hypothesize that this lacking association between fetal STS depth asymmetry and later language comprehension is related to the functional neural organization of language comprehension and not to the timeline of development.

A few associations have also been found between fetal STS depth asymmetry and later language-related white matter tracts. We included the superior longitudinal fasciculus, the inferior longitudinal fasciculus, the uncinate fasciculus, and the arcuate fasciculus in our analyses, as these white matter tracts are known to be part of the ventral and dorsal language streams[40]. Little hemispheric difference was found in fractional anisotropy indices, suggesting that this measurement is not ideal for laterality calculations. Analyses of tract volumes and the number of streams within the tracts revealed a considerable range of values and showed left lateralized indices for the arcuate fasciculus, the inferior longitudinal fasciculus, and superior longitudinal fasciculus I in the majority of children. However, while left superior longitudinal fasciculus III volume correlated with fetal STS

laterality, no laterality index of any tract was predicted by fetal STS depth asymmetry. Whereas brain-behavior relationships between white matter tract microstructures and cognitive functioning are well-known[41], laterality indices of different white matter measures have previously been shown to be an insensitive marker for language abilities and language lateralization[42]. They might therefore also fail to be associated with fetal STS depth asymmetry in the present study.

Importantly, as the association between fetal STS depth asymmetry and development observed in the present study is language-specific, no correlation was found between fetal sulcal asymmetry and nonverbal perceptual reasoning. This finding stresses the importance and specificity of this temporal region for early language development and contributes important information about the neuronal underpinnings of the development of language during the prenatal period.

A strength of this preregistered study is the prospective study design and the predefined study purpose, which focuses on the prognostic value of fetal STS asymmetry for language development: no post hoc analyses were conducted, and all measurement results planned for the study are reported in this paper. While the long time interval between fetal MRI and examinations of language abilities, language localization, and language-related white matter tracts is a further asset of the study, this study design caused several problems and resulted in some limitations. Fetal MRI is only performed with sufficient suspicion of abnormal fetal development and in former years, the overall number and imaging quality of fetal MRI examinations were lower than today. Since our goal was to trace the development of healthy fetuses, we had to go back several years in the fetal MRI databases to find enough potential study participants. This resulted in a relatively large participants' age range at test (6–13 years of age). Furthermore, nine study participants had to be excluded from the final statistical analyses as they had no visible STS in their fetal MRI. In hindsight, it would have been better to only include fetuses older than 24 weeks of gestation at MRI and children with visible STS in the follow-up investigations. As a consequence, the sample size of the present study is rather small and gestational at fetal MRIs varies considerably. As the MR research device became defective during the study, some study participants were measured with a different MR device. To minimize the influence of this change on the results, we adjusted all analyses involving MR data for age at examination and device and used age-corrected z-scores in all analyses of cognitive data. We thus aim to present this study as a pilot study which need to be replicated in a larger sample of healthy children as well as in different clinical populations.

Larger study groups will also need to investigate if the variance of fetal morphometric STS measures increases with gestational age as the brains individualize. It is a concern that possible non-linear growth effects and increases in normal variance of measures with gestational age may not have been adequately accounted for in the present study due to the small number of participants. The fact that the dataset is so small may mean that these more complex effects could not be detected in this small sample but could still impact the findings.

Similarly, the imbalance of sexes in our study sample is a limitation of this study. However, it reflects the clinical situation at fetal MRI diagnostics, where male fetuses are referred to more often than female fetuses due to a higher vulnerability and a higher incidence of prenatal damages[43]. Several studies have furthermore outlined that brain development may follow different pathways over time in boys and girls[44,45]. In addition, in the first years of life, girls show an advantage in language development compared to boys[46]. On the other hand, a comprehensive literature review on sex differences in brain and language development[47] reveals contradictory findings and limited evidence for sex differences in brain structures and functions associated with language processing. Furthermore, the asymmetry of the planum temporale has shown not to be significantly influenced by sex[24]. Thus, sex differences may play a role during certain developmental stages while they are negligible in other stages. In the present study, we did neither find significant differences between sexes in the fetal measurements nor in later language abilities; however, much larger studies are needed with smaller age ranges to investigate possible sex differences in the relationship between fetal STS depths and later language abilities.

The present study investigated children with a considerable variability in language functioning; however, no child had a neurological disease nor a language impairment. Structural-functional correspondences may be different in children with neurological diseases compared to healthy children. If the findings of the present study hold true in different patient groups, the fetal STS asymmetry has the potential to become a diagnostic biomarker of the maturity and integrity of neural correlates of language.

## Methods

**Participants**. The radiological database of the Department of Biomedical Imaging and Image-guided Therapy, Medical University of Vienna was searched for children whose mothers were transferred to fetal MRI diagnostics due to various clinical reasons 6–13 years ago and whose fetal MRIs were subsequently diagnosed as normal. The image quality from found fetal MRIs was first visually assessed, and then the images were reviewed again by an experienced neuroradiologist for the absence of structural brain anomalies. In sum, 155 children were identified as suitable for this study. We contacted these families to invite them to participate in our study and to ensure the following further inclusion criteria in the child: (a) normal or corrected-to-normal vision and normal hearing; (b) no history of neurological or psychiatric disease; (c) no general MR contraindication. Furthermore, both parents had to be native German speakers. Finally, 38 children and their parents matched the inclusion criteria and agreed to participate. In all of these children, a neuropsychological examination, a functional magnetic resonance imaging (fMRI) for language localization, and diffusion tensor imaging (DTI) to identify language-related white matter bundles were performed.

All children received a 30 € voucher for a bookstore. The study was registered prior to study start in the German Clinical Trials Register (DRKS00010582) and approved by the Ethics Committee of the Medical University of Vienna (1083/2015) in accordance with the Helsinki Declaration of 1975. The children were provided with age-appropriate assent forms and parents received a parental permission form. All children and one parent per child gave written, informed consent prior to inclusion.

**Fetal MRI acquisition and analysis**. Six to 13 years prior to the start of the study, fetal MRI examinations were performed using a 1.5 T scanner (Ingenia, Philips Medical Systems) in accordance with the ISUOG practice guidelines[48]. A body coil had been used and T2-weighted turbo spin-echo sequences (in-plane resolution, 0.62/0.62–1.17/1.17 mm; slice thickness, 2.0–4.5 mm; matrix size, 256 × 256; field of view, 200–230 mm; relaxation time, ≤20,000 ms; echo time, 100–140 ms) had been acquired in the three orthogonal planes.

At study, an experienced neuroradiologist (G.K.) and a neuroradiologist in training (P.K.) who were blinded for cognitive results measured the maximum depth of the fetal STS as initially described in Kasprian et al.[11]. For this purpose, a line conjoining the vertex of the forming superior temporal gyrus and the forming middle temporal gyrus was drawn. Sulcal depth of both STS was determined by measuring the distance between the deepest point of the STS and the baseline (for more detailed information, see Kasprian et al.[11]). For more accurate definition of the sulcal pit of the STS, super-resolution fetal MRI data was used. To determine if there was agreement between the two raters on STS depths measurements, ICC estimates and their 95% confident intervals were calculated based on absolute-agreement and a two-way mixed-effects model. There was excellent agreement between the two raters' measurements with an average measure of ICC = 0.998 with a 95% confidence interval from 0.993 to 0.999 (F = 409.19, $p < 0.001$). Examples of fetal images can be found in the Supplementary Figs. 7 and 8.

A laterality index of the fetal STS depths was calculated using the formula LI = [($\sum$ left $- \sum$ right)/($\sum$ left $+ \sum$ right)]*100. Thus, an LI value of 100 represents complete left lateralization, an LI value of $-100$ corresponds to complete right hemisphere dominance of STS depth. LI was categorized as left lateralized if ≥20, bilateral if within $-20$ and $+20$, or right if ≤ $-20$. In addition, an alternative asymmetry measure was calculated accounting for possible effects of brain volume and gestational age at fetal MRI, following the approach of Bullmore et al.[49] (please find a more detailed description of the method and the results of this

alternative approach in the Supplementary Note 1, Supplementary Table 6 and Supplementary Table 7).

**FMRI acquisition and analysis**. For this study, a magnetic resonance imaging (MRI) was performed to investigate functional language localization and lateralization. The first participants were scanned on a 3 T Siemens TIM Trio whole-body MR-Tomograph (Siemens Medical Solutions, Erlangen Germany). 3D structural MRI scans were performed using an isovoxel magnetization-prepared rapid gradient-echo (MPRAGE, T1-weighted, TE/TR_4.21/2300 ms, inversion time 900 ms, with a matrix size of 240 × 256 × 160, voxel size 1 × 1 × 1.10 mm, flip angle 9°) sequence. FMRI was acquired using a phase-corrected blipped gradient echo, single-shot echo-planar imaging (EPI) sequence. Altogether, 200 EPI volumes were acquired with a square FOV of 210 mm, voxel size 2.1 × 2.1 × 4 mm, 20 slices with a gap of 25 percent were aligned parallel to the AC-PC plane; repetition time (TR) was 2000 ms, echo time (TE) 42 ms, and the flip angle was set to 90 degrees. During the course of this study, the MR research scanner, unfortunately, became defective. Thus, 13/38 (7/29) children were scanned on a Philips Elition 3 Tesla scanner (EPI: TR 1000 ms, TE 25 ms, single-echo EPI, slice thickness 2.5 mm, flip angle 65°, 51 slices, 400 volumes; anatomical images: 0.75 × 0.75 × 1, flip angle 8°, TE 3.75 ms, TR 8.188 ms). To control for these differences, we included scanner type as a nuisance variable in the model for all statistic analyses.

The German version of an auditory description definition task was administered during fMRI assessment. Detailed description of this paradigm can be found in Bartha-Doering et al.[50], Bartha-Doering et al.[51], or Bartha-Doering et al.[52]. In the auditory description definition condition, the participants heard the definition of an object followed by a noun and were instructed to press a button each time the definition described the noun correctly. The control condition consisted of reverse speech, with some items additionally containing a pure tone at the end. Here, the participants are instructed to press the button each time he/she heard the tone. Three age-adjusted versions of the fMRI paradigm were available. The total fMRI scan time was 6 min and 40 s.

The images were pre-processed using Statistical Parametric Mapping 12 software (SPM; Wellcome Department of Cognitive Neurology, London, United Kingdom) working on MATLAB 2019a. EPI volumes were spatially realigned and corrected for movement. Frame to frame displacement between successive volumes was estimated by calculating the Euclidian distance from the translational parameters obtained from the realignment. Customized prior probability maps and a customized T1 template, matched to age and sex composition of the study group, were created by employing the Template-O-Matic (TOM) toolbox[53]. After co-registration, the derived spatial normalization parameters were used to normalize the functional volumes. Normalized EPI volumes were visually inspected for maximum overlap with the template and then smoothed using a spatial filter kernel of FWHM = 4 mm.

BOLD signal increases pertaining to task-evoked responses in brain activity were modeled using a general linear model as implemented in SPM. Six regressors modeling residual movement-related variance (translational and rotational movement) were included in the model as covariates of no-interest. Language activation was measured by contrasting the auditory description definition task condition with the reversed language control condition. Individual lateralization of activations was estimated at the single-subject level using the LI-toolbox[54]. LIs were computed both for the whole brain (LI global) and for the regions of interest (LI frontal, LI temporal, LI parietal). In order to avoid the threshold dependency of LIs, a bootstrapping approach was employed. A group map of language activation was derived using a one-sample t-test with an extended cluster size of 20. The association of language activations with the individual LIs of the fetal STS depths was calculated using multiple regression analysis, adjusted for age at test and MR device. Both second level analyses (t-test, multiple regression) were corrected for multiple comparisons family-wise error (FWE) with $p < 0.05$.

**DTI acquisition and analysis**. Structural T1-weighted MRI were processed using Freesurfer[55] to yield both an accurate segmentation of the cortical ribbon as well as tissue-type classification of every voxel in cortical gray matter, white matter, deep gray matter, and cerebro-spinal fluid. Echo planar Diffusion Weighted Images (DWI) were acquired during the same scanning session using the MR scanners as described above (b-values of 0 and 1000 s/mm²; 30 gradient encoding directions; acquired voxel size 2 × 2 × 2 mm; TR = 8000 ms; TE = 83 ms).

Pre-processing consisted of denoising using overcomplete local PCA[56], followed by skull-stripping and bias-field correction. Local orientation of myelinated fibers was estimated by constrained spherical deconvolution of the pre-processed DWI data using Mrtrix3[57]. Briefly, this process consists of estimating tissue-specific response functions which are used to compute a white matter fiber Orientation Distribution Function (fODF) that captures the structural arrangement of white matter at every location in the DWI data. A two-step procedure was performed to reduce the effects of signal distortion on later measurements. This procedure consisted in first estimating single-tissue response functions for WM and CSF to establish a first estimate of the fODF. From this, we generated an image of the Apparent Diffusion Coefficient, which has its highest values in locations filled with fluid eg. The CSF. We used this map to compute a rigid alignment between the previously estimated binary labels of CSF and the structural T1 and T2 images. Applying the resulting transformation, we can perform an approximate mapping of

the T1-weighted image into the DWI space. This in turn allows us to generate a pseudo-T1-weighted image from the DWI data. We then used Deformable registration via attribute matching and mutual-saliency weighting[58] to perform non-rigid alignment of this pseudo-T1-weighted image defined in DWI space and the actual, undistorted T1-weigthed image, thereby removing the geometric distortion present in the DWI data.

We followed Smith et al.[59] and performed whole-brain Anatomically Constrained Tractography (ACT) by dynamically generating 10 million individual streamlines per case. In addition to ACT, streamline seeding and filtering were performed as in Smith et al.[60] using the "Spherical-deconvolution informed filtering of tractograms 2" (SIFT2) method. The resulting whole-brain tractogram accurately reflects the underlying fiber structure and is less biased by random effects emanating from the sampling procedure. An example of a DTI measurement can be found in the Supplementary Fig. 9.

Further, we virtually dissected individual fiber bundles of interest (AF, SLF I, SLF II, SLF III, UF, ILF) from the whole-brain tractograms using the White Matter Query Language (WMQL)[61] in an automated, unbiased manner. We recorded the number of streamlines assigned to each fiber bundle, the corresponding volume of white matter as well as the average Fractional Anisotropy (FA) along all streamlines assigned to each individual tract to quantify the structure of each bundle. Laterality Indices (LIs) of the white matter tracts were calculated by using the formula LI = [(∑ left − ∑ right)/(∑ left + ∑ right)]*100. LI was categorized as left lateralized if ≥20, bilateral if within −20 and +20, or right if ≤−20.

**Examination and analysis of cognitive abilities**. Standardized tests of language comprehension, naming, verbal fluency, and verbal short- and long-term memory were used to assess verbal abilities. *Language comprehension* was measured with the Token Test for Children[62], where tokens varying in size and shape have to be moved according to auditory commands with increasing length and linguistic complexity. *Expressive vocabulary* was examined using the Wortschatz- und Wortfindungstest[63]. This test provides information about expressive vocabulary in different lexical categories, including nouns, verbs, and adverbs/adjectives. Semantic and phonemic *verbal fluency* were evaluated using the Regensburger Wortflüssigkeitstest,[64] which requires the participant to name, within 2 min, as many words as possible of the semantic category animals and of words starting with the letter "S", respectively. *Verbal memory* was assessed with the German version of the Auditory Verbal Learning Test[65], the Verbaler Lern- und Merkfähigkeitstest[66]. This test requires the child lo learn an aurally presented word list and measures short-term recall after distraction, long-term recall, and recognition of learned words.

As an estimation of nonverbal intelligence, the Perceptual Reasoning Index was derived from three subtests of the WISC IV[67]: The Mosaiktest (Block Design subtest) asks the child to recreate a design from a picture using colored blocks and requires visual processing of spatial relations and mental manipulations of visual patterns. In the Matritzentest (Matrix Reasoning subtest), the child needs to select a picture that completes a matrix; this subtest examines nonverbal problem solving, rules, and generalization. The Bilder Ergänzen Test (Picture Completion subtest) requires the child to detect missing parts from pictures and thus measures visual understanding and organization.

Raw scores of cognitive tests were transformed into age-adjusted z-scores for each test. For tests with more than one score, a mean z-score was calculated: for verbal fluency, a mean z-score was derived from the z-scores of the phonemic and semantic fluency subtests, for the verbal memory test, a mean z-score was calculated using the z-scores of short-term recall, long-term recall, and recognition. In line with clinical conventions, individual z-scores from −1 to 1 were defined as average. Performance above 1 was read as above average, results below −1 was read as below average, and z-scores below −2 were interpreted as reduced. Reduced scores in two language abilities would suggest a language impairment.

**Further questionnaires**. Handedness was evaluated with the Edinburgh Hand-edness Inventory scale[68] with inverted direction to match the other laterality indices, thus ranging from 1 (completely left-handed) to −1 (completely right-handed). Educational levels of parents and household income were used as indicators of the children's socioeconomic background. Educational levels were rated on a 7-point scale for the mother and the father separately, and the family household monthly gross net income was classified on an 11-point scale and rescaled to a 7-point scale. The socioeconomic status was calculated by taking the arithmetic mean of maternal education, paternal education, and household income (see Bartha-Doering et al.[69] for more information on the questions and analyses).

**Statistics and reproducibility**. Statistical analyses were conducted using IBM SPSS Statistics (Version 28). Cognitive data, LIs of the fetal STS depths, and DTI measurements were normally distributed as shown by the Kolmogorov-Smirnov Test for Normality (each $p > 0.05$). FMRI LIs did not follow a normal distribution (LI total D(23) = 0.31, $p < 0.001$). Group comparisons between normally distributed variables were performed with Student's t-test and paired sample t-tests were used to compare STS measurements between hemispheres. Correlations between normally distributed data were performed with Pearson's correlation. For correlations involving fMRI LIs, Spearman rank correlations were used.

Significance of findings was set based on a Bonferroni correction factor, i.e., $\alpha = 0.05$/number of comparisons. Multiple linear regression analyses were employed to examine the association of the LI of the fetal STS depths with z-scores of language functions, adjusted for nonverbal perceptual reasoning, and with DTI measurements, adjusted for age at test and MR device. The sample sizes were more than 22 and are given in each analysis, figure, and table.

**Reporting summary**. Further information on research design is available in the Nature Research Reporting Summary linked to this article.

## Data availability
Data of the study are available at https://osf.io/surv5/.

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

## Acknowledgements

This work was supported by the Austrian Science Fund (FWF), Grant KLI544-B27. We like to thank Anna-Lisa Schuler for her help in the examination of study participants.

## Author contributions

L.B.-D. conceptualization, methodology, data curation, data analysis, visualization, writing, funding acquisition, coordination, project administration; K.K. methodology, data acquisition, writing; E.S. data analysis, visualization, writing; F.PhS.F. data analysis, visualization, writing; G.L. conceptualization methodology, data analysis, writing; M.W., data acquisition, analysis, writing; S.L.-S. data acquisition, data analysis, writing; P.K. data acquisition, analysis, writing; M.S. data acquisition, analysis, writing; A.T. data acquisition, analysis, writing; S.M. data analysis, writing; J.A. data acquisition, data analysis, writing; D.P. supervision, methodology, data acquisition, writing; R.S. conceptualization methodology, investigation, data analysis, supervision, writing; G.K. conceptualization, methodology, data acquisition, data analysis, writing, supervision.

## Competing interests

The authors declare no competing interests.
