## [Peer Review File · Communications Biology]

Reviewers' comments:

Reviewer #1 (Remarks to the Author):

This paper presents preliminary results that explore a unique new area of research that seeks to link early fetal markers of brain development with later developmental outcomes in childhood. The primary contribution here is forming a link between markers of early fetal brain asymmetry in brain sulcation with measures of language development. As such, this finding would be of significant interest and importance to the brain research community and the findings would have impact both in understanding normal development and in developing new biomarkers for cases of potential clinical abnormalities during fetal growth. This group are in a strong position to address these questions having compiled a unique long term imaging database starting with fetal imaging. The work is also based on techniques for fetal imaging and fetal morphometric measures that have been carefully developed by the group over many years. Because of the potentially highly impactful findings, it is important to help the authors ensure they have not missed anything in the way the data was collected, selected or analyzed that might potentially allow false positive results to occur.

Overall, beyond the very interesting topic and background of the work, I feel there are some limitations to this new work and the way it is presented that raise concerns about the confidence in the reported findings: These overall concerns arise because the findings are based on both a small and a relatively heterogeneous dataset. Specifically, the 29 fetal measurements are made over a very wide range of gestational ages (>10 weeks). The corresponding language tests they are related to are additionally carried out over a wide range of childhood ages. There is very limited information about the distribution of the measures across these ages, or how the statistical modelling is carried out to address the age range: An important step in improving the paper would be to include a model equation in the methods section that captures how all covariates were incorporated and accounted for in the statistical testing.

The approach to statistical modelling is important because firstly fetal morphometric measures (for example linear distance or volume) over a large age range may be expected to increase in a non-linear fashion during growth. In addition, variance of basic morphometric measures would also be expected to increase with gestational age as the brains individualize (the authors speak to this in their statements that fetal measures potentially provide a clearer marker than those made postnatally). It is a concern that non-linear growth effects and increases in normal variance of measures with age may not have been adequately accounted for in the statistical models and tests. The fact that the dataset is so small may mean that these more complex effects cannot be detected in this small sample but could still impact the findings with a small fraction of (for example early or late) measures in this subsample driving an artifactual relationship.

A second concern on analysis is the nature of the variance in the ratio measures used for asymmetry, which would potentially not be normally distributed (the measure is apparently a ratio of possibly normally distributed measures). As with other aspects, it is a concern that the small sample size may mask the impact of these variances on the analysis. There are other alternative laterality measures and statistical approaches to asymmetry that may address the variances more appropriately.

An additional concern about the cohort is the significant imbalance of male and female fetuses in the study (10/19) that does not appear to be accounted for in the analysis. This is important because fetal brain morphometric measures and language scores may potentially vary between male and female subjects.

Another concern about the cohort is that it was apparently not based on simply healthy pregnancies, but fetuses that were initially scanned for clinical evaluations. No information is given about what the original reasons were for these clinical scans. This concern is further amplified by the absence of any baseline measures for normal postnatal development of the subjects (eg Bayley scores carried out at a

consistent postnatal age).

The graphs in Figure 1 present a convincing view of basic relationships but hides the potential impact of covariates. These should be augmented by including graphs of potential covariate relationships. This should include for example: graphs of LI measures against gestational age, LI measures against age at language test, language test score against age at language test and language test score against gestational age at fetal measurement. These plots should also be color labeled for male/female to highlight any imbalances with gender.

Another area of concern is that the basic fetal imaging measures are derived from manual marking by one or two observers. From the reference given for the methods, the variance in the manually derived measures of sulcal depth may be significant (Figure 4 in the supplied reference shows significant variability in sulcal depth increases with age in a much larger cohort). In this new smaller study (based on a subsample of this earlier data), it is therefore not clear how reproducible the asymmetry measures are for the data presented, either in terms of within or between observer consistency. More information on this process is needed to help interpret the significance of the results in relation to measurement errors. It is also not clear how the subsample of the earlier study was selected for this study, or if the observers making fetal measurements for this study were blinded to the language outcomes and other subject variables. The fetal imaging used to measure sulcal depth also did not use a consistent image resolution (slice thickness apparently varying between 2.0 and 4.5mm and in plane resolution varying from 0.62 to 1.17mm). It is a concern that this may also impact the accuracy of results given the sulcal depth differences from left to right are on the order of 2 or 3 mm. I would also suggest including some images of the fetal measurements made in the paper with voxel dimension markers and measurement scales (rather than images of postnatal data that do not relate to the primary findings in fetuses).

In relation to clarity in the paper: Much of the methods text is devoted to presenting and describing the analysis and results of postnatal imaging findings from fMRI and DTI, which are of interest but are not well linked to the key findings of this paper: particularly since these results are based on different subject cohorts to the actual fetal study results described in the title and abstract. Removing these altogether from the paper, or at least redoing the analysis on the same subjects as the fetal data would simplify the paper and interpretation of results. It is an additional concern that these postnatal studies were done on two very different scanners.

Overall, although this is a potentially highly impactful area of work, it is disappointing that such a small and heterogeneous dataset was used for this study, particularly given the earlier findings of the group on fetal brain asymmetry were derived from much larger cohorts. To avoid the chance of reporting false positives it may be better to wait until more of the fetal sample used in the earlier publications have had follow up language tests to allow a more focused and balanced study cohort which pose less challenges to statistical modelling.

Reviewer #2 (Remarks to the Author):

The present paper assessed, in a very rare longitudinal data sample, correlations between fetal anatomical characteristics of the superior temporal sulcus (STS) with verbal abilities and connectivity of the STS 6-13 years later. The authors show that a less rightward lateralization of the STS at fetal stage (i.e. the normal asymmetry observed in adult) is associated with better verbal abilities in their subjects 6-13 years later. The authors conclude that this pattern may promote the development of the left STS in utero, and consequently the development of language substrates and language abilities. This is a very interesting paper providing important novel hypothesis regarding the ontogeny of language. I only have minor comments:

1. It is unclear to me why children of all ages (6-13) participating in the fMRI and behavioral testing (including verbal ones) were pooled together. Can the authors demonstrate that their ability in all the

testing performed were not statistically different? If different, can the authors identify whether their main results were already observable at age 6 or are they still evolving later?

2. I understand that fetal MRI were done with a 1.5T and that the quality of images are probably not excellent but that would be of interest to add a figure with some brain examples showing the STS.

3. A figure showing the various tracts (e.g. AF, SLF) from DTI in 6-13 years-old participants would also be of interest.

4. p12, line 348: what does mean "MRT"

Dear reviewers,

We would like to thank for your valuable comments. Both reviewers offered concrete and very helpful suggestions.

We responded to each of your comment and suggestion in turn, with page and line numbers referring to the revised manuscript. Changes in the manuscript are highlighted in yellow.

We are convinced that the review process improved the manuscript significantly. We thank for your consideration and look forward to your response.

Lisa Bartha-Doering, in behalf of the authors

Reviewer #1

This paper presents preliminary results that explore a unique new area of research that seeks to link early fetal markers of brain development with later developmental outcomes in childhood. The primary contribution here is forming a link between markers of early fetal brain asymmetry in brain sulcation with measures of language development. As such, this finding would be of significant interest and importance to the brain research community and the findings would have impact both in understanding normal development and in developing new biomarkers for cases of potential clinical abnormalities during fetal growth. This group are in a strong position to address these questions having compiled a unique long term imaging database starting with fetal imaging. The work is also based on techniques for fetal imaging and fetal morphometric measures that have been carefully developed by the group over many years. Because of the potentially highly impactful findings, it is important to help the authors ensure they have not missed anything in the way the data was collected, selected or analyzed that might potentially allow false positive results to occur.

Reviewer #1 pt1: Overall, beyond the very interesting topic and background of the work, I feel there are some limitations to this new work and the way it is presented that raise concerns about the confidence in the reported findings: These overall concerns arise because the findings are based on both a small and a relatively heterogeneous dataset. Specifically, the 29 fetal measurements are made over a very wide range of gestational ages (>10 weeks). The corresponding language tests they are related to are additionally carried out over a wide range of childhood ages. There is very limited information about the distribution of the measures across these ages, or how the statistical modelling is carried out to address the age range: An important step in improving the paper would be to include a model equation in the methods section that captures how all covariates were incorporated and accounted for in the statistical testing.

The approach to statistical modelling is important because firstly fetal morphometric measures (for example linear distance or volume) over a large age range may be expected to increase in a non-linear fashion during growth. In addition, variance of basic morphometric measures would also be expected to increase with gestational age as the brains individualize (the authors speak to this in their statements that fetal measures potentially provide a clearer marker than those made postnatally). It is a concern that non-linear growth effects and increases in normal variance of measures with age may not have been adequately accounted for in the statistical models and tests. The fact that the dataset is

so small may mean that these more complex effects cannot be detected in this small sample but could still impact the findings with a small fraction of (for example early or late) measures in this subsample driving an artifactual relationship.

Authors: We thank for the positive comments and the constructive suggestions. We very much appreciate the reviewer's effort to help us improve our manuscript. We hope that the changes we made improved the manuscript and satisfy the reviewers.

Ad heterogeneity of gestational age at fetal MRI: We completely agree with the reviewer that there is some heterogeneity in the gestational age at fetal MRI. However, Austrian ethics regulations do not allow to perform fetal imaging for study purpose only. Thus, we could only include children which fetal MRIs were performed due to clinical reasons and which suspected diagnoses were not confirmed. For this reason, the setup of an exact timepoint of fetal imaging was not possible. We added this limitation in the discussion section (please see below), but nevertheless would like to present the reviewer and the readers alike that there is no correlation between gestational age at fetal MRI and laterality index of the STS depths in our sample. While the individual depths of the left and right hemisphere STS, respectively, significantly correlated with the gestational age at fetal MRI indicating larger bilateral STS depths in older fetuses, the laterality index of the STS depths was not associated with increasing age at fetal MRI in our study cohort with fetal MRI examinations between 25 and 35 gestational weeks. During this time frame, the visualization of the STS in both hemispheres is excellent and hemispheric differences are very accentuated (even by the unaided eye of an expert examiner). We totally agree with the reviewer that there may be differences in the laterality index at later stages of development (around birth or during the first postnatal years) – however, using our quite narrow time frame of 10 gestational weeks, those could not be observed.

We have had included an analysis on these parameters in the manuscript but we feel now that the way we presented the large amount of results was confusing. We therefore rearranged the whole results section and now describe results of each variable first, followed by the regression analyses between these variables. We hope that this improves the comprehensibility of the results section.

On page 4, we present the results of the correlation between gestational age at fetal MRI and STS depths:

“The individual depths of the left and right hemisphere STS significantly correlated with the gestational age at the time of the fetal MRI (right STS: $r = 0.645$, $p < .001$; left STS: $r = 0.677$, $p < .001$), indicating larger STS depth in older fetuses. However, the gestational age at fetal MRI did not correlate with the LI of the fetal STS depth ($r = .12$, $p = .551$), thus, increasing gestational age at fetal MRI was not associated with a significant change in the asymmetry of the fetal STS depth.” (p.5, lines 3-7)

We furthermore included a scatterplot to display the lack of correlation between gestational age at fetal MRI and fetal STS laterality index in the Appendix, page 4 (Figure A1). In accordance with the changes Reviewer#1 suggested in pt5, plots were color labeled for male/female. We incorporated this figure in a larger one with information on several covariate relationships, as suggested by Reviewer #1 in pt5.

“Figure A1. Covariate relationships. A) Correlation analysis between gestational age at fetal MRI and LI of the fetal STS depths shows no significant association between these variables ($r = .12$, $p = .551$). Thus, increasing gestational age at fetal MRI was not associated with a significant change in the asymmetry of the fetal STS depth. B) Similarly, age at test is not significantly associated with the LI of the STS depths at fetal MRI ($r = .15$, $p = .452$). C – F) In addition, age at test is not significantly associated with language abilities (all $p > .05$).” (Appendix, p.3)

Ad heterogeneity of age at test: We agree with the reviewer that the age range at test is also not ideal (please see the limitation paragraph below and the discussion where we discuss this point). To approach this problem, we only used standardized test and transformed all raw scores of the cognitive tests into age-corrected z-scores. This gives us information about the deviation of the individual performances in relation to the normative means of the respective age group (please see Methods section, p.16, for a detailed description). Thus, the cognitive outcome values are age-corrected. We nevertheless included an analysis of the association of age-corrected cognitive z-scores with age at test and did not find a significant correlation (please see Table A2 in the Appendix for detailed information). We mentioned this analysis also in the text on p. 5:

“Language scores did not significantly correlate with any background variables, including age at test, handedness, and socioeconomic background (Appendix, Table A2).” (p. 6, lines 9-10)

This age-correction, however, was not possible for fMRI and DTI data. We therefore adjusted the second level multiple regression analyses of fMRI language activations for age at test (and MRI device used). Similarly, the multiple linear regression to test the association of fetal STS depth asymmetry and later language-related white matter tracts was adjusted for age at test (and MRI device used).

We have realized that in the relevant descriptions of methods and results, we did not explicitly state “adjusted for age at test” but only wrote “adjusted for age”. This is of course confusing, as it could be mixed up with age at fetal MRI; we apologize for this mistake and corrected the sentences accordingly:

Results:

“However, second level multiple regression analyses of fMRI language activations, analyzed with SPM and adjusted for age at test and MR device used, revealed a significant association of the LIs of the fetal STS depth with activations in the left superior temporal lobe (Figure 2).” (p. 9, lines 5-8)

“Multiple linear regressions, adjusted for age at test and MR device used, were carried out to test if the LI of fetal STS depth predicted the lateralization of language-related white matter tracts 6-13 years later.” (p. 9, lines 17-19)

Methods:

“The association of language activations with the individual LIs of the fetal STS depths was calculated using multiple regression analysis, adjusted for age at test and MR device.” (p. 16, lines 29-31)

“Multiple linear regression analyses were employed to examine the association of the LI of the fetal STS depths with z-scores of language functions, adjusted for nonverbal perceptual reasoning, and with DTI measurements, adjusted for age at test and MR device.” (p. 19, lines 23-25)

We completely agree with Reviewer #1 that the small sample size and heterogeneity of data are significant limitations. We would like to convey the results of this study as preliminary results and as a pilot study which needs to be followed by larger ones, including studies with clinical populations. We

have mentioned these limitations in the Discussion section. In the revised version, we explicitly stressed the large age ranges and the pilot character of this study:

“A strength of this preregistered study is the prospective study design and the predefined study purpose, which focuses on the prognostic value of fetal STS asymmetry for language development: no post-hoc analyses were conducted, and all measurement results planned for the study are reported in this paper. While the long time interval between fetal MRI and examinations of language abilities, language localization, and language-related white matter tracts is a further asset of the study, this study design caused several problems and resulted in some limitations. Fetal MRI is only performed with sufficient suspicion of abnormal fetal development and in former years, the overall number and imaging quality of fetal MRI examinations were lower than today. Since our goal was to trace the development of healthy fetuses, we had to go back several years in the fetal MRI databases to find enough potential study participants. This resulted in a relatively large participants’ age range at test (6-13 years of age). Furthermore, nine study participants had to be excluded from the final statistical analyses as they had no visible STS in their fetal MRI. In hindsight, it would have been better to only include fetuses older than 24 weeks of gestation at MRI and children with visible STS in the follow-up investigations. As a consequence, the sample size of the present study is rather small and gestational at fetal MRIs varies considerably. Lastly, as the MR research device became defective during the study, some study participants were measured with a different MR device. To minimize the influence of this change on the results, we adjusted all analyses involving MR data for age at examination and device and used age-corrected z-scores in all analyses of cognitive data. We thus aim to present this study as a pilot study which need to be replicated in a larger sample of healthy children as well as in different clinical populations.” (p. 12f, lines 28ff)

Ad a model equation that captures how all covariates were incorporated and accounted for in the statistical testing: We are again grateful for this suggestion and included model equations for all regression analyses in the results section in Table 2, page 8:

Table 2. Association between the LI of the STS depths and language abilities

Outcome	R*	p	Regression model equation
Expressive vocabulary	.43	.007**	y = .50 + .03x₁ + .60x₂
Language comprehension	.15	.476	y = .53 + .01x ₁ + .51x ₂
Verbal fluency	.53	.002**	y = .14 + .02x₁ + .30x₂
Verbal memory	.59	.000**	y = .36 + .02x₁ + .22x₂

Note: *correlation of LI of the STS depths and language abilities corrected for nonverbal perceptual reasoning; x₁, LI of the STS depth variable; x₂, nonverbal perceptual reasoning variable. Bold letters indicate p < .05, ** indicates significance after Bonferroni correction.

Ad nonlinear growth of fetal STS morphometry: We share the concerns of the reviewer that our sample is too small to detect possible nonlinear complex morphometric STS growth effects. With the present sample, we feel we cannot solve this problem but only point it out to the reader. We therefore added this limitation in the discussion section:

“Larger study groups will also need to investigate if the variance of fetal morphometric STS measures increases with gestational age as the brains individualize. It is a concern that possible non-linear growth effects and increases in normal variance of measures with gestational age may not have been adequately accounted for in the present study due to the small number of participants. The fact that the dataset is so small may mean that these more complex effects could not be detected in this small sample but could still impact the findings.” (p. 13, lines 14-19)

Reviewer #1 0 pt2: A second concern on analysis is the nature of the variance in the ratio measures used for asymmetry, which would potentially not be normally distributed (the measure is apparently a ratio of possibly normally distributed measures). As with other aspects, it is a concern that the small sample size may mask the impact of these variances on the analysis. There are other alternative laterality measures and statistical approaches to asymmetry that may address the variances more appropriately.

Authors: We are not quite sure if we understood the critics of Reviewer #1 right. The laterality indices of the fetal STS depth were normally distributed, as tested by the Kolmogorov-Smirnov Test (we have described this in the chapter “statistics and reproducibility” (p. 19, lines 15-17) and add a figure here, in case this is of interest).

We chose this laterality measure as it is independent of the gestational age at fetal MRI, as shown in Figure A1 and described in the results on page 5. This independence of age is a huge advantage of using this laterality index calculation instead of other measures, especially in our study with quite a large age range at fetal MRI. We thus do not see the advantage of using a different laterality measure. If we, however, missed the point of Reviewer #1 here, we apologize and ask for further explanation.

Reviewer #1 pt3: An additional concern about the cohort is the significant imbalance of male and female fetuses in the study (10/19) that does not appear to be accounted for in the analysis. This is important because fetal brain morphometric measures and language scores may potentially vary between male and female subjects.

Authors: The Reviewer is right. There is an imbalance of sexes in the whole clinical fetal MRI dataset at the Medical University of Vienna, as is in our final sample. In general, more male fetuses are referred to fetal MRI diagnostics than female ones. This observation is underlined by several studies and discussed in the literature (e.g., Kraemer, BMJ 2000). Nevertheless, this is a limitation of our study which we now mentioned and discussed in the Discussion section.

Statistical analyses showed that there is no significant difference between sexes in fetal STS measures, language abilities, language lateralization, or lateralization of white matter tracts (please see Table A2 for detailed results). However, the reviewer is right, statistical significance is (in such a small sample) not the only important information. We therefore included box plots of possible differences of the most important variables (STS LI, language outcome) between sexes in the Appendix (Figure A2) to add more visual information on distribution and differences between sexes.

Nevertheless, in larger study groups, not only sex differences between single variables, but also sex differences in the relationship between these variables could become visible. We cannot investigate this hypothesis in the small data set, larger sample sizes are needed for this. We therefore included this point in the limitation paragraph.

Figure A2. Sex differences. Girls and boys did not significantly differ in laterality indices of the STS depths nor in language scores (all $p > .05$), but the group of boys showed a larger data variability than the group of girls.

“Similarly, the imbalance of sexes in our study sample is a limitation of this study. However, it reflects the clinical situation at fetal MRI diagnostics, where male fetuses are referred to more often than female fetuses due to a higher vulnerability and a higher incidence of prenatal damages⁴³. Several studies have furthermore outlined that brain development may follow different pathways over time in boys and girls^{44, 45}. In addition, in the first years of life, girls show an advantage in language development compared to boys⁴⁶. On the other hand, a comprehensive literature review on sex differences in brain and language development⁴⁷ reveals contradictory

findings and limited evidence for sex differences in brain structures and functions associated with language processing. Furthermore, the asymmetry of the planum temporale has shown not to be significantly influenced by sex²⁴. Thus, sex differences may play a role during certain developmental stages while they are negligible in other stages. In the present study, we did neither find significant differences between sexes in the fetal measurements nor in later language abilities; however, much larger studies are needed with smaller age ranges to investigate possible sex differences in the relationship between fetal STS depths and later language abilities.” (p. 13, lines 20-33)

Reviewer #1 pt4: Another concern about the cohort is that it was apparently not based on simply healthy pregnancies, but fetuses that were initially scanned for clinical evaluations. No information is given about what the original reasons were for these clinical scans. This concern is further amplified by the absence of any baseline measures for normal postnatal development of the subjects (eg Bayley scores carried out at a consistent postnatal age).

Authors: We are grateful for these important questions and included the reasons for referral to fetal MRI diagnostics on page 4. Unfortunately, we do not have any Bayley scale measurements after birth; however, besides a normal fetal MRI, all children had a normal structural MRI at test (ages 6-13 years) as well as a normal nonverbal IQ and no language disorder at test. We therefore consider the study participants typically developed children. We added the point that all children had normal structural MRI findings at test in the Results section to be more clear here:

“Reasons for referral to fetal MRI have been either diseases of the mother including metabolic diseases (n = 9), amniotic fluid problems (n = 3), tumours (n = 2), and elevated risks of abortion (e.g., uterus rupture in previous pregnancy, previous abortion, n = 6), or suspected diseases of the fetus including urogenital abnormalities (e.g., kidney cysts, hydronephrosis, n = 8), gastrointestinal abnormalities (e.g., diaphragmatic hernia, gastrochisis, n = 6), club foot (n = 2), and brain abnormalities (1 plexus cyst, 1 mild ventriculomegaly).” (p. 4, lines 6-11)

“Fetal structural MRI and structural MRI at test revealed normal findings in all children.” (p. 4, lines 14-15)

Reviewer #1 pt5: The graphs in Figure 1 present a convincing view of basic relationships but hides the potential impact of covariates. These should be augmented by including graphs of potential covariate relationships. This should include for example: graphs of LI measures against gestational age, LI measures against age at language test, language test score against age at language test and language test score against gestational age at fetal measurement. These plots should also be color labeled for male/female to highlight any imbalances with gender.

Authors: We thank Reviewer#1 for this suggestion and included the suggested graphs in the Appendix, page 3 (please see also our answer to Reviewer #1 pt1 on page 3 of this document with the included figure.

For each regression analysis, we furthermore included a statistical analysis of multicollinearity of the independent variables as well as a Durbin-Watson (DW) statistics to investigate autocorrelation in residuals. We added the variance inflation factor (VIF) and the DW in the Results.

“Tests to see if the data met the assumption of collinearity indicated that multicollinearity was not a concern (LI of STS depths, Tolerance = 1.00, VIF = 1.00; nonverbal perceptual reasoning, Tolerance = 1.00, VIF = 1.00). Furthermore, the data met the assumption of independent errors (expressive vocabulary, Durbin-Watson value = 1.74; verbal flexibility, Durbin-Watson value = 1.54; verbal memory, Durbin-Watson value = 1.94; language comprehension, Durbin-Watson value = 1.93).” (p. 7, lines 25-30)

Reviewer #1 pt6: Another area of concern is that the basic fetal imaging measures are derived from manual marking by one or two observers. From the reference given for the methods, the variance in the manually derived measures of sulcal depth may be significant (Figure 4 in the supplied reference shows significant variability in sulcal depth increases with age in a much larger cohort). In this new smaller study (based on a subsample of this earlier data), it is therefore not clear how reproducible the asymmetry measures are for the data presented, either in terms of within or between observer consistency. More information on this process is needed to help interpret the significance of the results in relation to measurement errors. It is also not clear how the subsample of the earlier study was selected for this study, or if the observers making fetal measurements for this study were blinded to the language outcomes and other subject variables. The fetal imaging used to measure sulcal depth also did not use a consistent image resolution (slice thickness apparently varying between 2.0 and 4.5mm and in plane resolution varying from 0.62 to 1.17mm). It is a concern that this may also impact the accuracy of results given the sulcal depth differences from left to right are on the order of 2 or 3 mm. I would also suggest including some images of the fetal measurements made in the paper with voxel dimension markers and measurement scales (rather than images of postnatal data that do not relate to the primary findings in fetuses).

Authors: We thank the reviewer for these important points of criticism. We now describe the process of fetal imaging measures in more detail. We also added some images of the fetal measurements in the Appendix (Figures A7 and A8) as suggested. Furthermore, we do agree that manually measuring the depths of the STS may harbor certain errors. That’s why we are now providing statistical data of the interrater agreement of STS measurements that show that the depth determinations in our study are robust and reliable.

“At study, an experienced neuroradiologist (G.K.) and a neuroradiologist in training (P.K.) who were blinded for cognitive results measured the maximum depth of the fetal STS as initially described in Kasprian, *et al.*¹¹.” (p. 15, lines 4-6)

“To determine if there was agreement between the two raters on STS depths measurements, ICC estimates and their 95% confident intervals were calculated based on absolute-agreement and a two-way mixed-effects model. There was excellent agreement between the two raters’ measurements with an average measure of ICC = .998 with a 95% confidence interval from .993 to .999 (F= 409.19, p < .001).” (p. 15, lines 10-14)

“Examples of fetal images can be found in the Appendix, Figures A7 and A8.” (p. 15, lines 14-15)

Figure A7. Coronal slices of superresolution-reconstructed fetal brains. A. Fetal brain at GA 21+1. STS are not developed on both sides; temporal lobes appear symmetrical. B: Typical asymmetry pattern at GA 28+3. The arrows point towards the STS, the right one is more deeply developed.

Figure A8. Comparison of two fetuses both at GA 28+3, performed with a respective field strength of 1.5 (A) and 3 Tesla (B).

Reviewer # 1 pt7: In relation to clarity in the paper: Much of the methods text is devoted to presenting and describing the analysis and results of postnatal imaging findings from fMRI and DTI, which are of interest but are not well linked to the key findings of this paper: particularly since these results are based on different subject cohorts to the actual fetal study results described in the title and abstract. Removing these altogether from the paper, or at least redoing the analysis on the same subjects as the fetal data would simplify the paper and interpretation of results. It is an additional concern that these postnatal studies were done on two very different scanners.

Authors: We are grateful for the hint that our data presentation was quite confusing. Actually, all statistical analyses of fetal STS depths, fMRI, DTI, and language evaluations were only done within the group of 29 children with visible STS. We recognized that we were not clear enough about this. To improve the clarity of the manuscript, we rearranged the Results section and now present the analyses in the way we have performed them: first the data of the whole group of participants (n = 38) with details in the Appendix, then the data of the 29 children with visible fetal STS (text and Table 1), followed by the results of each variable, and finally, the associations between variables. We furthermore added the number of participants in the description of results of each variable. Please see below the changes in the text we have done to improve clarity:

“The overall study group consisted of 38 children (age 6-13 years, mean 8.85, SD 1.98, 14 girls) whose mothers were transferred to fetal MRI diagnostics due to various clinical reasons and whose fetal MRIs were subsequently diagnosed as normal. [...] Thus, 29 study participants were included in the quantitative analyses of the fetal STS and in all further statistical analyses. [...] Within the study group with visible fetal STS (n = 29; age 6-13 years, mean 9.19, SD 2.03, 10 girls; Table 1), the mean depth of the right STS was significantly larger than its left counterpart (t = 6.494, p < .001).” (p. 4, lines 4-[...] 23).

“Language testing revealed typically developed verbal abilities in 24/29 children.” (p. 6, line 2)

“The functional magnetic resonance imaging (fMRI) measurement of one child (m) had to be aborted due to claustrophobia, and measurements of five children (3m, 2 f) had to be excluded due to excessive movement related artifacts. Head movement of the remaining 23/29 children was within the tolerable limits.” (p. 6, lines 15-18)

“Data on white matter tracts are available in 25/29 children.” (p. 7, line 2)

The reviewer is also right about the limitation regarding the two MRI devices. There is no way to change this, but we have adjusted the regression analyses for the MR device used. We mentioned this limitation in our discussion:

“As the MR research device became defective during the study, some study participants were measured with a different MR device. To minimize the influence of this change on the results, we adjusted all analyses involving MR data for age at examination and device and used age-corrected z-scores in all analyses of cognitive data. We thus aim to present this study as a pilot study which need to be replicated in a larger sample of healthy children as well as in different clinical populations.” (p. 13, lines 8-13)

Reviewer #1 pt8: Overall, although this is a potentially highly impactful area of work, it is disappointing that such a small and heterogeneous dataset was used for this study, particularly given the earlier findings of the group on fetal brain asymmetry were derived from much larger cohorts. To avoid the chance of reporting false positives it may be better to wait until more of the fetal sample used in the earlier publications have had follow up language tests to allow a more focused and balanced study cohort which pose less challenges to statistical modelling.

Authors: We agree that a larger sample size would increase the significance of findings. Further, we share the reviewers' disappointment concerning sample size and heterogeneity. Again, we would like to emphasize our experience in performing such a complex and long-term follow-up study being challenged by recalling subjects to complex follow-up testing at timepoints far distant to the original event (= imaging before birth). Given these experience and limitations, we believe that our dataset is still remarkable. Unfortunately, funding of this project has ended, and further MRI and cognitive measurements are not possible without. As we are one of the largest fetal MRI centers worldwide, leading to high number of subjects being imaged prenatally – we are positive to be able to provide less heterogeneous and larger study cohorts in future. Thus, we are preparing a follow-up application for

funding where we would like to go a step further, i.e. including children with neurological disorders, but also including additional healthy participants.

Reviewer #2:

The present paper assessed, in a very rare longitudinal data sample, correlations between fetal anatomical characteristics of the superior temporal sulcus (STS) with verbal abilities and connectivity of the STS 6-13 years later. The authors show that a less rightward lateralization of the STS at fetal stage (i.e. the normal asymmetry observed in adult) is associated with better verbal abilities in their subjects 6-13 years later. The authors conclude that this pattern may promote the development of the left STS in utero, and consequently the development of language substrates and language abilities. This is a very interesting paper providing important novel hypothesis regarding the ontogeny of language. I only have minor comments:

Reviewer #2 pt1: It is unclear to me why children of all ages (6-13) participating in the fMRI and behavioral testing (including verbal ones) were pooled together. Can the authors demonstrate that their ability in all the testing performed were not statistically different? If different, can the authors identify whether their main results were already observable at age 6 or are they still evolving later?

Authors: We are not quite sure if we understand the question right, we do want to apologize in advance if we misunderstood this point. Does Reviewer#2 mean that we pooled together all fMRI measurements? This is not what we did. For the functional MRI paradigm, three versions with different difficulty levels according to the age of the participant were available. Previous studies of our group have already shown that this paradigm with these different age-adjusted versions elicits robust language lateralization in children aged 6-16 (e.g., Bartha-Doering et al., Brain Lang 2018; Brain Behav 2018, EJP 2019; Dev Sci 2021). We then analyzed the fMRI measurements of each child individually and calculated a laterality index for each child. Statistics showed that individual functional MRI laterality indices did not significantly correlate with age at test (please see Appendix, Table A2 for detailed results). So, fMRI laterality indices were not age-dependent.

For cognitive measurements, age-corrected z-scores were used to inform about the deviation of the individual performances in relation to the normative means of the respective age group. Consequently, age-corrected z-scores of language abilities did not significantly correlate with age at test (Table A2). Please see also our answer and revisions to Reviewer#1 pt1. We hope that this answers the question of Reviewer#2 and that we can sufficiently demonstrate that the participants' language scores and laterality indices were not age-dependent.

Reviewer #2 pt2: I understand that fetal MRI were done with a 1.5T and that the quality of images are probably not excellent but that would be of interest to add a figure with some brain examples showing the STS.

Authors: We do agree that further image visualizations are important for presenting our data. Thus, we have now provided image examples of hemispheric asymmetries at 1.5 Tesla (Figure A7 in the Appendix). We have furthermore added a comparison of two fetuses at GA 28 imaged at 1.5 and 3 Tesla,

respectively, which demonstrate only a slight difference in the depiction of the STS at the studied fetal age (Figure A8). The slightly improved resolution of 3 Tesla is thus often traded against its susceptibility to motion. In our setting this would have led to an even higher dropout rate.

Reviewer #2 pt3: A figure showing the various tracts (e.g. AF, SLF) from DTI in 6-13 years-old participants would also be of interest.

Authors: We are again happy to provide this figure (Figure A9 in the Appendix) and added this information in the text:

“An example of a DTI measurement can be found in the Appendix, Figure A9.” (p. 17, line 29)

Figure A9. Example of DTI measures in a study participant (f, 12 years old). Arcuate fasciculus, red; inferior longitudinal fasciculus, yellow; superior longitudinal fasciculus I, pink; superior longitudinal fasciculus II, purple; superior longitudinal fasciculus III, orange; uncinatus fasciculus, green.

Reviewer #2 pt4: p12, line 348: what does mean "MRT"

Authors: We thank for this hint and corrected the mistake:

“For this study, a magnetic resonance imaging (MRI) was performed to investigate functional language localization and lateralization.” (p. 15, lines 22-32)

Authors: Additional changes that were made:

- Dr. Michael Weber was added as coauthor. Dr. Weber is an experienced statistician who helped with all the statistical methods questions and changes.
- We included subheadings in the Appendix to facilitate the reading.
- Figure and table legends now include sample sizes and statistics, as required by the formatting guidelines of the journal.

Reviewers' comments:

Reviewer #1 (Remarks to the Author):

This is a novel study using a valuable data set that allows unique exploration of long term development after early in-utero measures. The results are certainly potentially very interesting early findings.

Overall, the authors have responded well to the concerns raised in the first round of review and their modifications to the paper help reflect both the significance and potential limitations of the study.

However, given the importance of the work there are some residual concerns that I feel have not yet been addressed adequately. Specifically, perhaps due to my own unclear communication in the review, one important concern is the use of only the laterality index to assess asymmetry:

I was concerned about use of this measure given the availability of potentially more appropriate statistical approaches to asymmetry that also account for overall growth/head size effects. One concern is that LI is a ratio of two measures (length measurements) that themselves might be reasonably assumed to have normally distributed errors. The result of taking a ratio of these normally distributed variables would be expected to follow a Cauchy distribution, see for example:

https://en.wikipedia.org/wiki/Ratio_distribution

I am concerned that the use of a K-S normality test presented on the small sample of LI was not adequate to support the normality assertion, see for example:

Ghasemi A, Zahediasl S. Normality tests for statistical analysis: a guide for non-statisticians. *Int J Endocrinol Metab.* 2012 Spring;10(2):486-9. doi: 10.5812/ijem.3505. Epub 2012 Apr 20. PMID: 23843808; PMCID: PMC3693611.

who discuss the K-S test limitations on small samples. Further issues with the LI measure such as the different impact of the denominator components in the ratio are also important: This problem with the LI laterality index and alternative approaches have been discussed in the literature, for example for volume measures in:

Bullmore E, Brammer M, Harvey I, Ron M. Against the laterality index as a measure of cerebral asymmetry. *Psychiatry Res.* 1995 Aug 8;61(2):121-4. doi: 10.1016/0925-4927(95)02618-8. PMID: 7480389.

This paper recommends a more direct regression modeling approach to assessing asymmetry, that takes into account the overall growth/head size variations (potentially correlating with age) that the authors are rightly concerned about.

It would seem reasonable, given the potential issues with the laterality ratio measure combined with the very small dataset, that the authors could include analysis with alternative asymmetry measures (such as the regression modelling in the paper above), in addition to the laterality index measure to further strengthen the claims made from the small sample that is available.

A second concern is that the relatively preliminary nature of these findings are not conveyed in either the title or abstract. A few words concerning this would perhaps avoid over interpretation of the study results by those outside the field of fetal morphometry and bio-statistics.

Reviewer #2 (Remarks to the Author):

The authors responded to all my comments appropriately. As such I would recommend this article for publication.

Dear Editor, dear Reviewers,

We would like to thank you for your valuable comments. We understand Reviewer #1's remaining statistical concerns and followed the approach of Bullmore et al. (1995), as suggested by Reviewer #1, to investigate if age and brain volume had a different impact on the two measures (left and right STS depth). This analysis shows that left and right STS depths are symmetrically determined by gestational age and brain volume at fetal MRI. We nevertheless calculated an alternative lateralization score using a regression modeling approach, as suggested by the Editor and Reviewer #1. This new RDS score predicts language outcome to a similar extent as the LI score (please see the detailed description of analyses and results in the response to the Reviewer). Thus, as a) age and volume do not influence left and right measures to a different extent, and b) the alternative RDS score does not yield different results compared to the LI, we think there is no need to add the RDS score in the analyses of the manuscript. However, if the Editor and the Reviewer wish so, we are happy to add it. We furthermore updated the title and abstract, as suggested by Reviewer #1.

We are convinced that the overall review process improved the manuscript significantly. We thank for your consideration and look forward to your response.

Lisa Bartha-Doering, in behalf of the authors

Reviewer #1: *This is a novel study using a valuable data set that allows unique exploration of long term development after early in-utero measures. The results are certainly potentially very interesting early findings. Overall, the authors have responded well to the concerns raised in the first round of review and their modifications to the paper help reflect both the significance and potential limitations of the study. However, given the importance of the work there are some residual concerns that I feel have not yet been addressed adequately. Specifically, perhaps due to my own unclear communication in the review, one importantly concerns the use of only the laterality index to assess asymmetry: I was concerned about use of this measure given the availability of potentially more appropriate statistical approaches to asymmetry that also account for overall growth/head size effects. One concern is that LI is a ratio of two measures (length measurements) that themselves might be reasonably assumed to have normally distributed errors. The result of taking a ratio of these normally distributed variables would be expected to follow a Cauchy distribution, see for example https://en.wikipedia.org/wiki/Ratio_distribution. I am concerned that the use of a K-S normality test presented on the small sample of LI was not adequate to support the normality assertion, see for example: Ghasemi A, Zahediasl S. Normality tests for statistical analysis: a guide for non-statisticians. *Int J Endocrinol Metab.* 2012 Spring;10(2):486-9. doi: 10.5812/ijem.3505. Epub 2012 Apr 20. PMID: 23843808; PMCID: PMC3693611 who discuss the K-S test limitations on small samples. Further issues with the LI measure such as the different impact of the denominator components in the ratio are also important: This problem with the LI laterality index and alternative approaches have been discussed in the literature, for example for volume measures in: Bullmore E, Brammer M, Harvey I, Ron M. Against the laterality index as a measure of cerebral asymmetry. *Psychiatry Res.* 1995 Aug 8;61(2):121-4. doi: 10.1016/0925-4927(95)02618-8. PMID: 7480389. This paper recommends a more direct regression*

modeling approach to assessing asymmetry, that takes into account the overall growth/head size variations (potentially correlating with age) that the authors are rightly concerned about. It would seem reasonable, given the potential issues with the laterality ratio measure combined with the very small dataset, that the authors could include analysis with alternative asymmetry measures (such as the regression modelling in the paper above), in addition to the laterality index measure to further strengthen the claims made from the small sample that is available.

Authors: We thank the reviewer very much for this explanation and understand the point now. We don't have fetal data on head size, but we do have brain volume and gestational age at fetal MRI as variables that might influence left and right STS depths to a different degree.

We followed the approach of Bullmore et al. (1995), as suggested by Reviewer #1, and used multiple regression models to examine the effects of gestational age at fetal MRI and fetal brain volume on STS depths. Right and left STS depths were separately treated as dependent variables. The table shows estimated regression coefficients for the explanatory terms in each model and the results of the two-tailed t-tests of the null hypothesis that the true coefficients were zero.

Explanatory variable	Right STS depth		Left STS depth	
	β	p	β	p
Gestational age at fetal MRI	0.645	.000	0.677	.000
Brain volume at fetal MRI	0.643	.001	0.575	.004

Gestational age at fetal MRI and brain volume at fetal MRI both have a powerful and approximately symmetrical effect on left and right STS depths. Thus, left and right STS depths are symmetrically determined by gestational age and brain volume at fetal MRI. This observation suggests that the “common denominator” assumption implicit in the LI used in the present study was justified. We added this information in the manuscript on page 5:

“The individual depths of the left and right hemisphere STS significantly correlated with the gestational age at the time of the fetal MRI (right STS: $r = 0.645$, $p < .001$; left STS: $r = 0.677$, $p < .001$), indicating a symmetrical effect of gestational age on each side of STS depth and larger STS depths in older fetuses (Table A2).” (p. 5)

Nevertheless, we followed the approach of Bullmore et al. (1995) and calculated a residual difference score (RDS): $RDS = \text{residual right STS depths} - \text{residual left STS depth}$. Residual right and left STS depths were generated from regression models fitted to age at fetal MRI and brain volume. Multiple linear regressions were then carried out to test if the RDS of fetal STS depth predicted language abilities 6-13 years later. Analyses were adjusted for nonverbal perceptual reasoning. A comparison of these results with the multiple linear regression results involving the LI shows that both, RDS and LI, predict language outcome to a comparable degree:

Outcome	RDS		LI	
	R	p	R	p
Expressive vocabulary	.40	.021*	.43	.007**
Language comprehension	.15	.414	.15	.476
Verbal fluency	.60	.001**	.53	.002**
Verbal memory	.54	.004**	.59	.000**

For this reason, we did not include this alternative approach in the manuscript. However, if the Editor and Reviewer #1 think the RDS analysis adds new information that would be important for the manuscript, we would be happy to add it.

Reviewer #1: *A second concern is that the relatively preliminary nature of these findings are not conveyed in either the title or abstract. A few words concerning this would perhaps avoid over interpretation of the study results by those outside the field of fetal morphometry and bio-statistics.*

Authors: We agree with Reviewer#1 and changed the title accordingly. We furthermore revised the abstract and added a last sentence to underline the preliminary nature of our findings. As the word count limit for the abstract is 200, we had to shorten the previous abstract text a little:

Title: “The prognostic value of fetal temporal sulcus depth asymmetry for language development”

Abstract: “In most humans, the superior temporal sulcus (STS) shows a rightward depth asymmetry. This asymmetry can not only be observed in adults, but is already recognizable in the fetal brain. As the STS lies adjacent to brain areas important for language, STS depth asymmetry may represent an anatomical marker for language abilities. This study investigated the prognostic value of STS depth asymmetry in healthy fetuses for later language abilities, language localization, and language-related white matter tracts. Less right lateralization of the fetal STS depth was significantly associated with better verbal abilities, with fetal STS depth asymmetry explaining more than 40% of variance in verbal skills 6-13 years later. Furthermore, less right fetal STS depth asymmetry correlated with increased left language localization during childhood. We hypothesize that earlier and/or more localized fetal development of the left temporal cortex is accompanied by an earlier development of the left STS and is favorable for early language learning. If the findings of this pilot study hold true in larger samples of healthy children and in different clinical populations, fetal STS asymmetry has the potential to become a diagnostic biomarker of the maturity and integrity of neural correlates of language.” (p. 2)

REVIEWERS' COMMENTS:

Reviewer #1 (Remarks to the Author):

I thank the authors for doing the extra work and running this new analysis! The results seems to further support the findings and I think it is worth including both analysis approaches in the paper (particularly if readers are aware of the the other methods and may have similar concerns given the small sample size).

Other than this addition, I think the paper is ready for publication.

Dear Editor, dear Reviewer#1,

We would like to thank you for your valuable comments. According to the Reviewer #1's remaining suggestion, we included the alternative statistical approach to calculate STS asymmetry in the Supplementary file. We furthermore updated the title according to the editor's suggestion.

We are convinced that the overall review process improved the manuscript significantly. We thank for your consideration and look forward to your response.

Lisa Bartha-Doering, in behalf of the authors

Reviewer #1: *I thank the authors for doing the extra work and running this new analysis! The results seems to further support the findings and I think it is worth including both analysis approaches in the paper (particularly if readers are aware of the the other methods and may have similar concerns given the small sample size). Other than this addition, I think the paper is ready for publication.*

Authors: We have now included the alternative statistical approach to measure STS depth asymmetry in the Supplementary File and refer to it in the Methods section on page 13:

“Additionally, an alternative asymmetry measure was calculated accounting for possible effects of brain volume and gestational age at fetal MRI, following the approach of Bullmore et al. ⁴⁹ (please find a more detailed description of the method and the results of this alternative approach in the Supplementary Information, S5).” (p. 13)

Further changes:

As suggested in the “Final Revision Instructions”, we changed the title into “Fetal temporal sulcus depth asymmetry has prognostic value for language development” and changed the name “Appendix” into “Supplementary File”.